# The glutamate/cystine xCT antiporter antagonizes glutamine metabolism and reduces nutrient flexibility

Chun-Shik Shin[1], Prashant Mishra[1,2], Jeramie D. Watrous[3], Valerio Carelli[4,5], Marilena D'Aurelio[6], Mohit Jain[3] & David C. Chan[1]

As noted by Warburg, many cancer cells depend on the consumption of glucose. We performed a genetic screen to identify factors responsible for glucose addiction and recovered the two subunits of the xCT antiporter (system $x_c^-$), which plays an antioxidant role by exporting glutamate for cystine. Disruption of the xCT antiporter greatly improves cell viability after glucose withdrawal, because conservation of glutamate enables cells to maintain mitochondrial respiration. In some breast cancer cells, xCT antiporter expression is upregulated through the antioxidant transcription factor Nrf2 and contributes to their requirement for glucose as a carbon source. In cells carrying patient-derived mitochondrial DNA mutations, the xCT antiporter is upregulated and its inhibition improves mitochondrial function and cell viability. Therefore, although upregulation of the xCT antiporter promotes antioxidant defence, it antagonizes glutamine metabolism and restricts nutrient flexibility. In cells with mitochondrial dysfunction, the potential utility of xCT antiporter inhibition should be further tested.

[1] Division of Biology and Biological Engineering, California Institute of Technology, Pasadena, California 91125, USA. [2] Children's Medical Center Research Institute, University of Texas Southwestern Medical Center, Dallas, Texas 75390–8502, USA. [3] Department of Medicine and Pharmacology, University of California, La Jolla, San Diego, California 92093, USA. [4] IRCCS Istituto delle Scienze Neurologiche di Bologna, Bellaria Hospital, Via Altura 3, 40139 Bologna, Italy. [5] Department of Biomedical and NeuroMotor Sciences (DIBINEM), University of Bologna, Via Altura 3, 40139 Bologna, Italy. [6] Department of Neurology and Neuroscience, Weill Medical College of Cornell University, 1300 York Avenue, A501, New York, New York 10065, USA. Correspondence and requests for materials should be addressed to D.C.C. (email: dchan@caltech.edu).

For cultured mammalian cells, the two major carbon sources are glucose and glutamine. Catabolism of these two nutrients generates the majority of cellular energy, building blocks, and reducing equivalents for cell growth and proliferation. In rapidly growing cancer cells, these metabolic demands are accentuated, and oncogenesis often results in metabolic reprogramming to fuel the increase in cell biomass necessary for constant cell divisions[1–3]. In the Warburg effect, the most well studied form of metabolic reprogramming in cancer cells, aerobic glycolysis is used to consume large amounts of glucose with excess carbon secreted as lactate. This mode of metabolism persists despite high enough levels of oxygen to support oxidative phosphorylation (OXPHOS) in the mitochondria[1–3]. Metabolic reprogramming allows glucose to provide biosynthetic intermediates for the synthesis of proteins, lipids and nucleotides in rapidly proliferating cancer cells[4]. Many cancer cells also consume large amounts of glutamine, whose catabolism replenishes intermediates for the mitochondrial trichloroacetic acid (TCA) cycle (a process termed anaplerosis) and provides nitrogen for the synthesis of non-essential amino acids and nucleotides[5].

To what extent are glucose and glutamine interchangeable as carbon sources? In the absence of glucose, glutamine consumption in some cells is sufficient to protect cell viability[6–8]. This effect occurs via glutamine oxidation through the mitochondrial TCA cycle. However, some cancer cells have limited metabolic flexibility. First, the catabolism of glucose and glutamine in cancer cells can be specialized to provide distinct benefits to the cell. In proliferating glioblastoma cells, glucose metabolism is an important source for cellular lipids, whereas glutamine metabolism supports NADPH synthesis and replenishment of the TCA intermediate oxaloacetate[9]. Second, oncogenic reprogramming of metabolism can make cancer cells 'addicted' to either glucose or glutamine. Activation of the phosphoinositide 3-kinase (PI3K)-Akt pathway enhances glucose consumption and

glycolysis, and makes cancer cells highly susceptible to cell death following glucose withdrawal[10]. The proto-oncogene MYC stimulates glutamine metabolism and makes cells highly dependent on glutamine to prevent apoptosis[11,12]. In these cases, the rewiring of glucose or glutamine metabolism promotes rapid cell growth and division but limits flexibility in the use of alternative nutrients. Such metabolic reprogramming may therefore generate unique vulnerabilities that can be exploited for therapy[13].

There is little known about the factors that limit the nutrient flexibility of cells. To study this issue, we performed a genetic screen in human haploid cells to identify factors that constrain cells to utilization of glucose versus glutamine. We identified the SLC3A2 and SLC7A11 subunits of the xCT amino acid transporter (system $x_c^-$), which exports glutamate in exchange for cystine, a precursor for synthesis of the antioxidant glutathione. Downregulation of system $x_c^-$ function markedly improves cell viability under glucose-deficient/glutamine-replete conditions, due to enhanced ability to use intracellular glutamate to maintain respiratory chain activity. Furthermore, we identified Nrf2, an important transcription factor for the SLC7A11 gene, as a factor that limits the ability of breast cancer cells to utilize glutamine instead of glucose. In cybrid cells harbouring mitochondrial DNA (mtDNA) mutations, SLC7A11 is upregulated and its inhibition improves survival in galactose medium, where cellular bioenergetics rely primarily on mitochondrial OXPHOS through glutamine oxidation[14]. Our results show that system $x_c^-$, in addition to its well-known antioxidant role, is an important metabolic regulator that affects the nutrient flexibility of cells.

## Results

**A haploid genetic screen for glucose dependence.** Many immortalized cell lines show limited nutritional flexibility and are

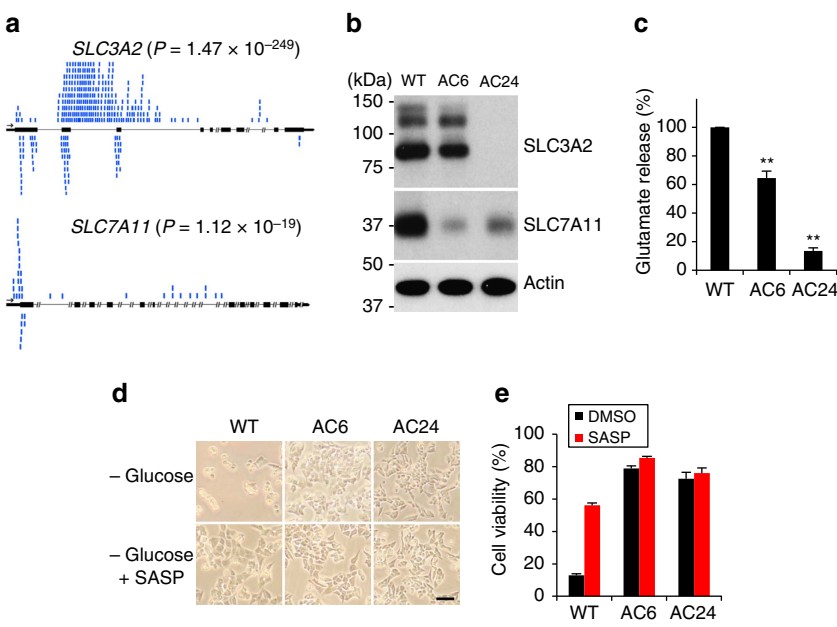

**Figure 1 | Identification of SLC3A2 or SLC7A11 as factors limiting cell viability under glucose-deficient conditions.** (a) Depiction of gene-trap insertions found in the SLC3A2 and SLC7A11 genes, in Hap1 cells surviving glucose depletion. Black rectangles represent exons, and blue marks represent insertion events. The number of gene-trap insertions from the unselected and selected populations was analysed by the one-sided Fisher exact test to calculate P values. (b) Western blot analysis of SLC3A2 and SLC7A11 levels in WT Hap1 (WT), AC6 (SLC7A11 gene-trap clone) and AC24 (SLC3A2 gene-trap clone) cells. (c) Quantification of glutamate release into the medium. Values were normalized to WT. Data represent the means ± s.d. (n = 3); **P < 0.01; unpaired Student's t-test. (d,e) Representative images (d) and quantification (e) of cell viability at 24 h after glucose withdrawal, with vehicle (DMSO) or 500 μM sulfasalazine (SASP). Scale bar, 200 μm. Data represent the means ± s.d. (n = 3).

highly dependent on glucose as the primary carbon source. We found that survival of the human haploid Hap1 cell line requires glucose in the culture medium. To identify factors involved in such 'glucose addiction', we performed a haploid genetic screen[15] to isolate mutants that survive in the complete absence of glucose. We randomly mutagenized $1 \times 10^8$ Hap1 cells with low multiplicity-of-infection with a retroviral gene trap vector[16] and cultured the mutagenized population in glucose-deficient medium for 12 days. After the majority ($>99\%$) of cells died, cells resistant to glucose depletion were recovered and expanded in nutrient-rich medium. Gene-trap insertion sites from the resistant population were identified using inverse-PCR-based Illumina sequencing[17]. In the selected population, the genes SLC3A2/4F2hc (399 distinct insertions) and xCT/SLC7A11 (39 insertions) were disrupted at high frequency by retroviral integration (Fig. 1a). Remarkably, the protein products of these genes are known to physically interact, with the SLC3A2 subunit termed the heavy chain and the SLC7A11 subunit termed the light chain, to form the amino acid antiporter known as system $x_c^-$ or the xCT antiporter[18]. The xCT antiporter is a cell-surface amino acid transporter that exports glutamate in exchange for cystine. This exchange is important for defence against cellular reactive oxygen species (ROS), because cystine is the oxidized dimer of cysteine, a critical precursor for synthesis of glutathione (GSH), a major antioxidant[19]. For clarity, we refer to the functional transporter as 'system $x_c^-$' or the 'xCT antiporter', and the light chain subunit as SLC7A11.

To characterize the cellular consequences of these gene disruptions, we isolated Hap1 clones with SLC3A2 and SLC7A11 gene-trap insertions. The SLC3A2 mutant, termed AC24, has a gene-trap insertion in the second intron of SLC3A2 in the expected orientation for gene disruption and shows no SLC3A2 expression. In addition, it has substantially lower SLC7A11 expression, which is expected because SLC3A2 must associate with SLC7A11 to form a stable complex (Fig. 1b). The SLC7A11 mutant, termed AC6, has a gene-trap insertion in the 5′ untranslated region of the first exon of SLC7A11 and shows greatly decreased SLC7A11 expression at the protein (Fig. 1b) and mRNA levels (Supplementary Fig. 1). SLC3A2 levels are largely unaffected in AC6, probably because the SLC3A2 subunit is shared by several amino acid transporters[18]. Both clones show lower glutamate release into the media (Fig. 1c), confirming that system $x_c^-$ is functionally disrupted. Importantly, both clones show highly improved viability under glucose-deficient conditions (Fig. 1d,e). After 24 h of glucose deprivation, $>70\%$ of the cells in both clones remain viable, whereas only 10% of parental Hap1 cells survive. Consistent with this result, treatment of Hap1 cells with sulfasalazine (SASP)[20], a system $x_c^-$ inhibitor, improved the viability of WT Hap1 cells after glucose withdrawal (Fig. 1d,e).

**System $x_c^-$ regulates cell viability during glucose withdrawal.** To independently confirm the results of the genetic screen, we used short hairpin RNAs (shRNAs) to stably knockdown system $x_c^-$ in Hap1 cells (Fig. 2a). Because SLC3A2 has pleiotropic functions as the common heavy chain for several amino acid transporters[18], we focused our knockdown efforts on the SLC7A11 subunit. Cells containing either of two independent SLC7A11 shRNAs released substantially less glutamate into medium (Fig. 2b). They showed greatly improved viability after glucose withdrawal (Fig. 2c). However, there was no change in proliferation rate in glucose-containing media (Supplementary Fig. 2a).

Conversely, we wondered whether increasing the level of system $x_c^-$ in a cell line with low endogenous SLC7A11 would sensitize cells to glucose withdrawal. Compared to Hap1 cells, HeLa cells expressed almost undetectable levels of the SLC7A11 subunit (Fig. 2a) and exhibited low glutamate release into the medium (Fig. 2b). Interestingly, control HeLa cells showed high viability in the face of glucose removal (Fig. 2d). Overexpression of the SLC7A11 subunit caused a large increase in glutamate release (Fig. 2a,b) and greatly increased cell death during glucose depletion (Fig. 2d). However, in glucose-containing media, there was no change in the proliferation rate of SLC7A11-over-expressing HeLa cells compared to the control (Supplementary Fig. 2a). Thus, overexpression of SLC7A11 is sufficient to elicit glucose addiction. Similar results were obtained using low glucose (versus no glucose in the experiments above) culture conditions (Supplementary Fig. 2b). In addition, neither knockdown nor overexpression of SLC7A11 significantly changed cell viability upon glutamine depletion (Supplementary Fig. 2c). Taken together, these results indicate that the level of system $x_c^-$ activity controls the sensitivity of cells to glucose withdrawal.

**The benefits of xCT depletion require glutamine metabolism.** Upon glucose withdrawal, glutamine becomes the primary carbon source and its metabolism is critical for cell survival[6–8]. Once imported, intracellular glutamine is converted into glutamate and further to α-ketoglutarate, an intermediate of the TCA cycle. Therefore, we speculated that cells with low system $x_c^-$ activity may be better able to maintain intracellular glutamate levels and its metabolism into the TCA cycle. To test this idea, we directly measured intracellular glutamate levels 1 h after glucose withdrawal. SLC7A11 knockdown cells (Fig. 3a) and SLC3A2 or SLC7A11 gene-trap mutants (Supplementary Fig. 3a) maintained their intracellular glutamate level after glucose depletion far better than control cells. Conversely, SLC7A11-overexpressing HeLa cells had lower levels of intracellular glutamate than control cells (Fig. 3b). Consistent with the known role of system $x_c^-$ for GSH synthesis, SLC7A11 knockdown reduced the level of intracellular GSH in Hap1 cells, whereas SLC7A11-overexpressing HeLa cells showed higher GSH levels than control cells (Supplementary Fig. 3b).

We next asked whether glutamine/glutamate metabolism is important for the enhanced cell survival of system $x_c^-$-deficient cells during glucose withdrawal. We tested three compounds [bis-2-(5-phenylacetamido-1,3,4-thiadiazol-2-yl)ethyl sulfide (BPTES, glutaminase GLS1 inhibitor], aminooxyacetate (AOA, amino-transferase inhibitor) and epigallocatechin gallate (EGCG, glutamate dehydrogenase inhibitor)[21] that inhibit enzymes involved in conversion of glutamine to glutamate and then α-ketoglutarate (Fig. 3c). Upon treatment with EGCG, Hap1 cells disrupted for SLC7A11 (Fig. 3d; Supplementary Fig. 3c) or SLC3A2 (Supplementary Fig. 3c) no longer show increased viability under glucose deprivation, suggesting that glutamate dehydrogenase (GDH) is critical for the effect. Because GDH converts glutamate to α-ketoglutarate, we asked whether α-ketoglutarate could reverse the inhibitory effect of EGCG. Indeed, the inhibitory effect of EGCG was fully rescued by supplementation with dimethyl α-ketoglutarate (dm-αKG), a cell-permeable form of α-ketoglutarate, (Fig. 3e; Supplementary Fig. 3d). Furthermore, dm-αKG supplementation during glucose withdrawal greatly improved cell viability in wild-type (WT) and control shRNA Hap1 cells (Fig. 3e; Supplementary Fig. 3d) and SLC7A11-overexpressing HeLa cells (Fig. 2d), whereas additional supplementation of glutamine into the medium did not significantly improve cell viability (Supplementary Fig. 3e). Therefore, these results suggest that conversion of glutamate to α-ketoglutarate underlies the protective effect of system $x_c^-$ disruption under glucose-deficient conditions. The lack of effect by BPTES or AOA suggests that GLS1 is not the primary

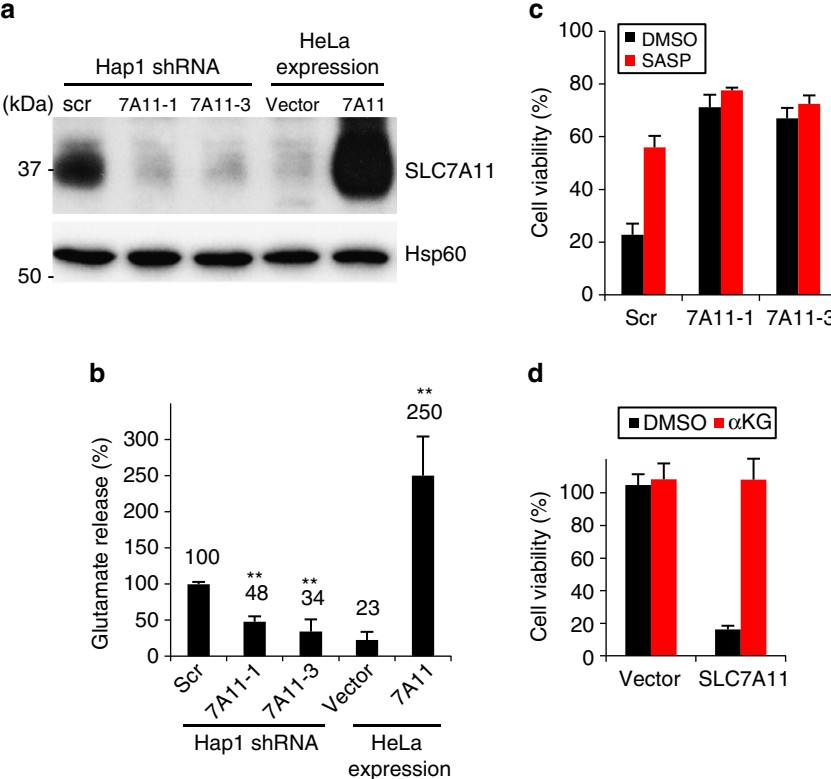

**Figure 2 | System x$_c^-$ negatively impacts Hap1 and HeLa cell viability after glucose withdrawal.** (**a**) Western blot analysis of SLC7A11 protein levels in *SLC7A11* knockdown Hap1 and *SLC7A11*-overexpressing HeLa cells. (**b**) Quantification of glutamate release into the medium. Data represent the means ± s.d. (*n* = 4); **P < 0.01; unpaired Student's *t*-test; scr, scrambled shRNA. (**c**,**d**) Cell viability at 24 h after glucose withdrawal. *SLC7A11* knockdown Hap1 cells (**c**) and *SLC7A11*-overexpressing HeLa cells (**d**) were cultured in glucose-deficient medium with vehicle (DMSO), 500 µM SASP or 4 mM dm-αKG for 24 h. Data represent the means ± s.d. (*n* = 3); vector: empty vector.

glutaminase in Hap1 cells, and that GDH, rather than aminotransferase, is important for conversion of glutamate to α-ketoglutarate.

In the absence of glucose, mitochondrial OXPHOS (as the major source of ATP synthesis) becomes essential for survival. Through anaplerosis, glutamine-derived α-ketoglutarate replenishes intermediates of the TCA cycle, which generates substrates to drive OXPHOS[21]. We therefore tested whether the levels of system x$_c^-$ can regulate mitochondrial respiration. In the presence of glucose, the oxygen consumption rate (OCR) of *SLC7A11* knockdown Hap1 or overexpressing HeLa cells is comparable with that of control cells (Supplementary Fig. 3f). However, *SLC7A11* knockdown cells show higher mitochondrial respiration after 3 h of glucose depletion, whereas the OCR of SLC7A11-overexpressing HeLa cells is lower than that of control cells under glucose withdrawal (Fig. 3f).

Recent work has found that α-ketoglutarate generated from consumed glutamine may undergo oxidative metabolism via the 'forward', clockwise TCA cycle or reductive carboxylation via a 'reverse' counter clockwise TCA reaction[22–24]. Relative flux through the forward versus reverse TCA cycle may be determined using stable $^{13}$C-isotope labelling (C5) of extracellular glutamine and measuring label incorporation into TCA cycle intermediates[22,23,25]. To determine whether modulation of system x$_c^-$ alters forward versus reverse TCA cycle flux, *SLC7A11* knockdown Hap1 or overexpressing HeLa cells were cultured in the absence of glucose and with U-$^{13}$C$_5$-glutamine (see Methods). Exposure to extracellular U-$^{13}$C$_5$-glutamine resulted in near complete labelling (M + 5) of intracellular glutamine and glutamate (Supplementary Fig. 3g). Labelled glutamate entered the TCA via anaplerotic reaction to

α-ketoglutarate with M + 4 labelling of the forward TCA intermediates succinate, fumarate, malate and citrate (Fig. 3g). The M + 5 label of citrate (reductive TCA product) was observed to a much lesser degree (∼8:1 M + 4:M + 5 citrate), indicative of low reverse TCA flux. Importantly, the relative degree of forward versus reverse flux was unaltered with either knockdown or overexpression of *SLC7A11*.

**xCT regulates dependence of breast cancer cells on glucose.** To extend our observations on the negative effect of system x$_c^-$ on glutamine metabolism and respiration, we analysed gene expression data for 947 cancer cell lines from the Cancer Cell Line Encyclopedia (CCLE) project[26]. For each cancer type, we categorized 15–20 cancer cell lines as high and low *SLC7A11* expressing and performed gene set enrichment analysis between these two groups. We found that genes related to OXPHOS were enriched in the low *SLC7A11*-expressing groups, especially in lung (P value = 0.018) and breast cancer cell lines (P value = 0.039). Moreover, analysis of 59 breast cancer cells showed a marked and selective negative correlation between the expression of *SLC7A11* and mitochondrial OXPHOS genes (Supplementary Fig. 4a,b). This observation suggests that most breast cancer cells with low *SLC7A11* expression have an expression profile consistent with upregulation of the OXPHOS machinery, and vice versa.

We chose two breast cancer cell lines, Hs578T and SK-BR-3, as representative of high and low *SLC7A11* expressing cell lines, respectively (Supplementary Fig. 4a). Consistent with the CCLE expression data, Hs578T cells showed a high expression level of SLC7A11 and active release of glutamate into medium, whereas

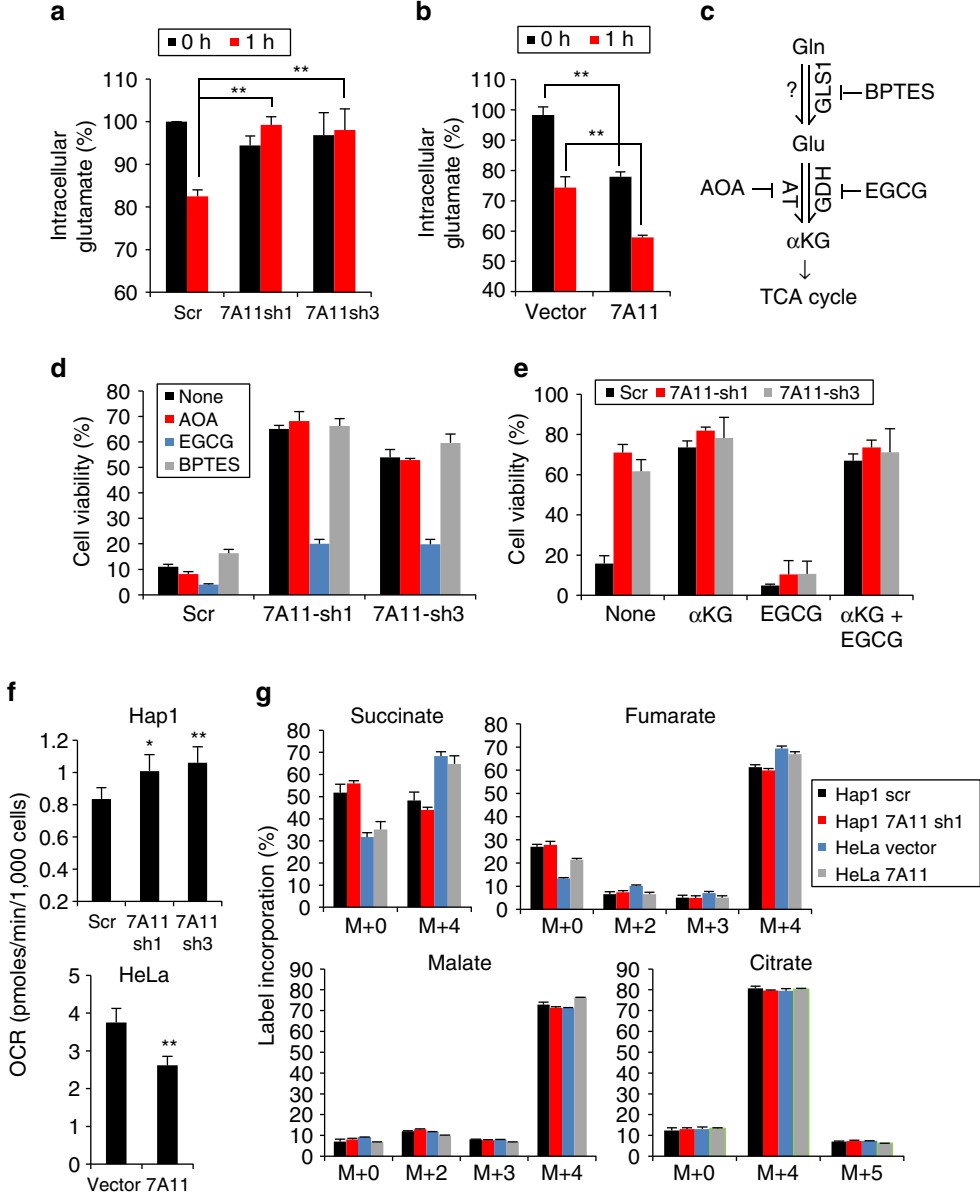

**Figure 3 | The pro-survival effects of *SLC7A11* depletion require glutamine metabolism.** (**a,b**) Quantification of intracellular glutamate in *SLC7A11* knockdown Hap1 cells (**a**) and *SLC7A11*-overexpressing HeLa cells (**b**) before and 1 h after glucose removal. Data represent the means ± s.d. ($n = 3$); \*\*$P < 0.01$; unpaired Student's *t*-test. (**c**) Enzymes and inhibitors involved in glutamine/glutamate metabolism. (**d,e**) Hap1 cell viability at 24 h after glucose withdrawal. The following drugs were added as indicated: 0.5 mM AOA, 50 μM EGCG, 10 μM BPTES or 4 mM dm-αKG. Data represent the means ± s.d. ($n = 3$). (**f**) Oxygen consumption under glucose-deplete conditions. Cells were incubated for 3 h in glucose-deficient medium (1 mM glutamine) and OCR was measured. Data represent the means ± s.d. ($n = 4$); \*$P < 0.05$. \*\*$P < 0.01$; unpaired Student's *t*-test. (**g**) Isotope labelling patterns of the TCA intermediates. Cells were incubated for 8 h in glucose-free media containing U-$^{13}$C$_5$-glutamine before extracting metabolites. Data represent the means ± s.d. ($n = 3$).

SK-BR-3 cells showed low SLC7A11 expression and no detectable release of glutamate (Fig. 4a; Supplementary Fig. 5a). We also determined the metabolic status of Hs578T and SK-BR-3 cells by measuring the activities of OCR and glycolysis (extracellular acidification rate (ECAR)). Hs578T cells showed lower OCR and a lower OCR/ECAR ratio than SK-BR-3 (Fig. 4b,c), suggesting that Hs578T cells are more dependent on glycolysis than SK-BR-3. Most Hs578T cells, like Hap1 cells, died upon glucose withdrawal (Fig. 4d), whereas SK-BR-3 cells were resistant (Fig. 4e). Hs578T cells depleted for *SLC7A11* showed decreased glutamate release into medium (Supplementary Fig. 5a) and more effectively maintained intracellular glutamate levels under glucose depletion (Supplementary Fig. 5b). Importantly, *SLC7A11*

knockdown greatly improved Hs578T cell viability during glucose withdrawal (Fig. 4d). However, Hs578T cells lacking SLC7A11 had higher cellular ROS levels than control cells, consistent with its role in antioxidant defence (Supplementary Fig. 5c). Conversely, cells overexpressing SLC7A11 had lower ROS levels (Supplementary Fig. 5c). The survival of *SLC7A11* knockdown Hs578T cells during glucose withdrawal was abolished by EGCG and BPTES, again implying that production of α-ketoglutarate from glutamate is critical (Fig. 4d). As expected, the detrimental effects of EGCG or BPTES treatment were fully reversed by dm-αKG supplementation (Fig. 4d). The differential effect of BPTES (compare Fig. 4d with Fig. 3d) implies that GLS1 is the primary glutaminase in Hs578T cells but not Hap1 cells.

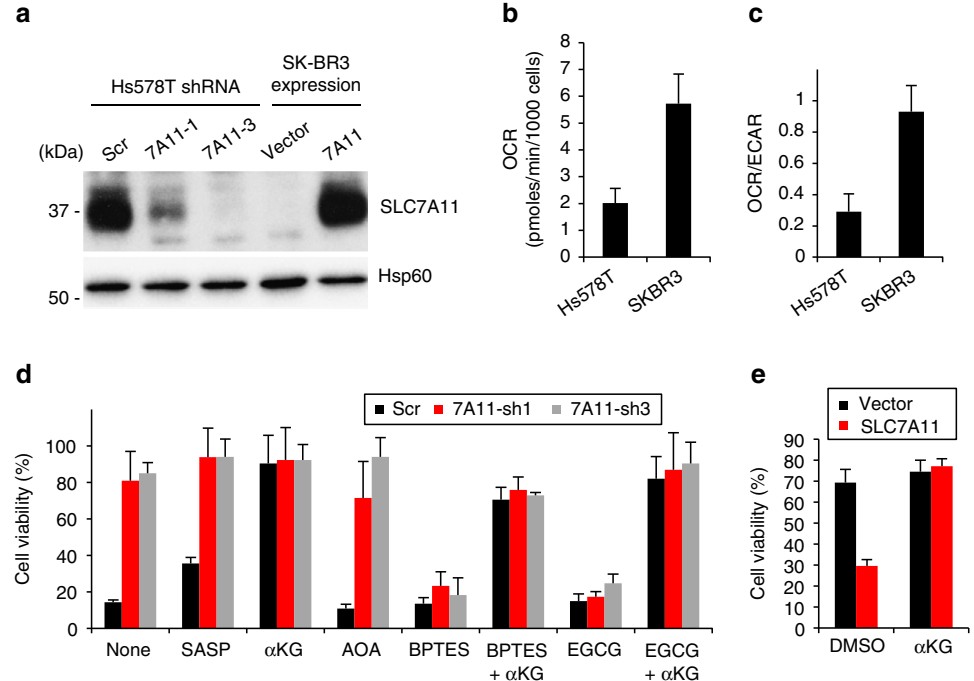

**Figure 4 | Depletion of SLC7A11 promotes survival of breast cancer cells after glucose depletion.** (**a**) SLC7A11 protein levels in *SLC7A11*-knockdown Hs578T and *SLC7A11*-overexpressing SK-BR-3 cells. (**b,c**) Metabolic status under glucose-replete conditions. The OCR (**b**) and OCR/ECAR ratio (**c**) were measured in the presence of 10 mM glucose + 2 mM glutamine. Data represent the means ± s.d. ($n = 3$). (**d,e**) Cell viability 24 h after glucose withdrawal. *SLC7A11* knockdown Hs578T (**d**) and *SLC7A11*-overexpressing SK-BR-3 cells (**e**) were cultured in glucose-deficient medium for 24 h. The following drugs were added as indicated: 500 μM SASP, 0.5 mM AOA, 50 μM EGCG, 10 μM BPTES or 4 mM dm-αKG. Data represent the means ± s.d. ($n = 3$).

Overexpression of SLC7A11 in SK-BR-3 cells increased glutamate release (Supplementary Fig. 5a) and concomitantly reduced the ability to maintain intracellular glutamate under glucose-deficient conditions (Supplementary Fig. 5b). SLC7A11 overexpression caused substantial SK-BR-3 cell death after glucose withdrawal. Cell death was rescued by addition of dm-αKG, suggesting that overexpression of SLC7A11 leads to a deficiency of α-ketoglutarate under glucose-deficient conditions (Fig. 4e).

Hs578T cells with *SLC7A11* knockdown were able to maintain their respiratory activity for prolonged periods after glucose withdrawal (Supplementary Fig. 5d). Conversely, SK-BR-3 cells overexpressing SLC7A11 showed a steep decline in OCR under the same regiment. Taken together, these results indicate that system $x_c^-$ in breast cancer cells regulates intracellular glutamate flux into the TCA cycle and increases sensitivity to glucose withdrawal.

**Nrf2 operates upstream of SLC7A11.** Nrf2 (nuclear factor erythroid 2-like 2, NFE2L2) is a key transcription factor that upregulates many cytoprotective genes involved in oxidative stress[27]. Because *SLC7A11* is a known target gene of Nrf2 (ref. 27), we examined the correlation of *Nrf2* and *SLC7A11* expression in cancer cells. *Nrf2* expression is positively correlated with *SLC7A11* expression across 59 breast cancer cell lines ($R = 0.5688$), as well as all 947 cancer cell lines ($R = 0.4306$) in the CCLE expression data. Moreover, *Nrf2* target gene expression shows high correlation with *SLC7A11* expression across 947 cancer cell lines. The correlation coefficient ($R$) between *SLC7A11* and *NRF2* target genes are: *NQO1* (+ 0.4179), *GCLC* (+ 0.3283), *GCLM* (+ 0.3944) and *TXNRD1* (+ 0.4321).

These correlations suggest that Nrf2 supports *SLC7A11* expression in a subset of cancer cells. To test this idea, we knocked down *Nrf2* in Hs578T and MDA-MB-231 breast cancer

cells, two high expressors of *SLC7A11*. *Nrf2* knockdown caused a marked decrease in *SLC7A11* expression (Fig. 5a,b,e) and reduced glutamate release (Fig. 5c,f). Importantly, *Nrf2*-depleted cells showed greatly improved viability during glucose withdrawal (Fig. 5d,g). Again, this change in viability is dependent on glutamate metabolism, because treatment with EGCG abolished the pro-survival effect of *Nrf2* knockdown (Fig. 5d). In addition, α-ketoglutarate supplementation was sufficient to promote cell survival, even in the presence of EGCG.

To examine the effect of Nrf2 upregulation, we treated cells with dimethyl fumarate (DMF), a cell-permeable Nrf2 activator[28]. DMF treatment caused MDA-MB-231 cells to highly induce *SLC7A11* expression and glutamate release (Fig. 5e,f). Moreover, cells pretreated with DMF showed substantially reduced viability during glucose depletion compared to control cells (Fig. 5g). We ruled out off-target effects by showing that these outcomes of DMF treatment were dependent on both SLC7A11 and Nrf2 (Fig. 5f,g). Taken together, these results indicate that the Nrf2–SLC7A11 axis regulates glutamate metabolism, the dependence of cancer cells on glucose, and their ability to utilize glutamine as an alternative carbon source. Importantly, it suggests that adaptation to oxidative stress in cancer cells can compromise nutrient flexibility, particularly through the enhancement of a glucose addiction phenotype.

**xCT regulates mitochondrial function in mtDNA-mutant cells.** Our findings above indicate that cells with system $x_c^-$ upregulation due to oxidative stress may have restricted nutrient flexibility due to impaired glutamine metabolism. Inhibition of the mitochondrial respiratory chain can increase reactive oxygen species[29,30]. We found that SLC7A11 expression was induced by drugs that block mitochondrial function, especially rotenone and oligomycin, which inhibit complex I (NADH dehydrogenase) and V (ATP synthase) of the OXPHOS machinery, respectively

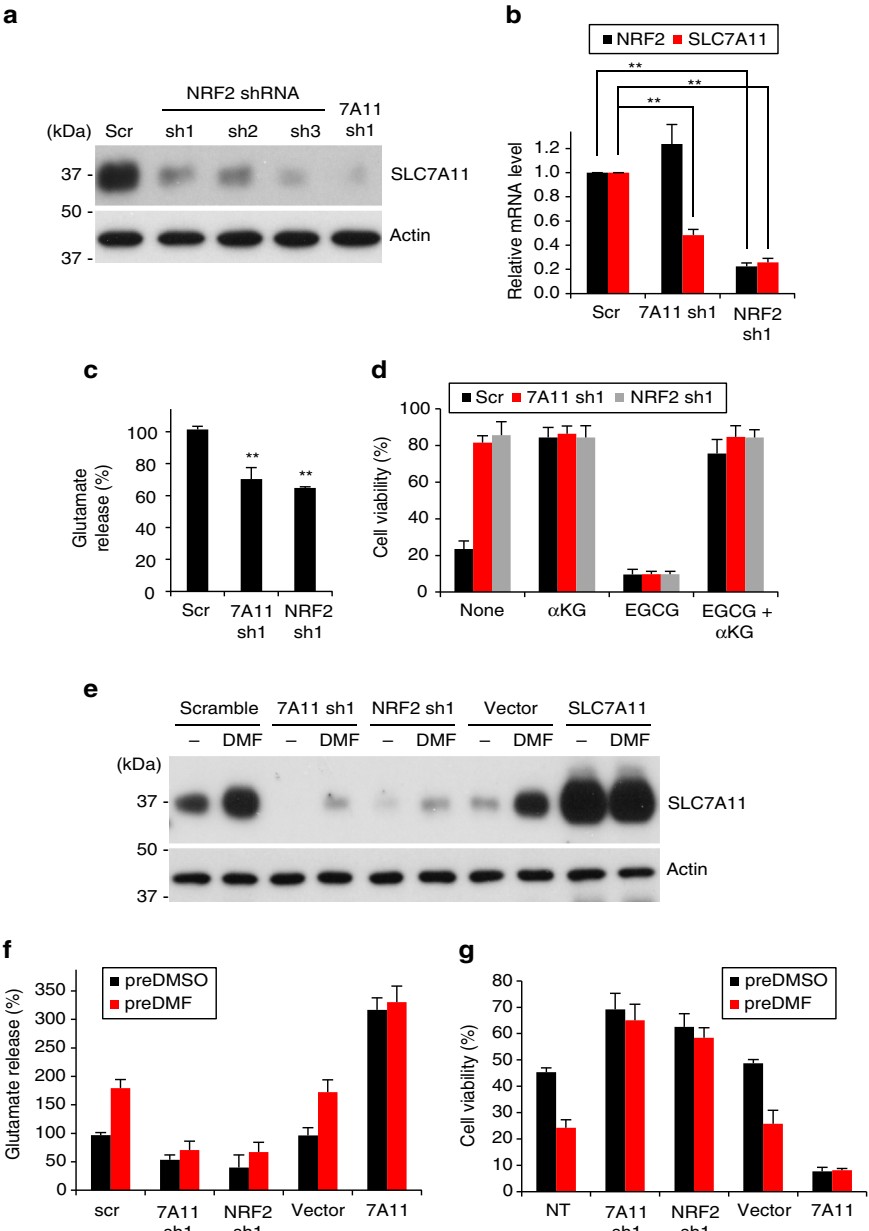

**Figure 5 | Nrf2 regulates *SLC7A11* expression and survival of breast cancer cells upon glucose withdrawal.** (**a**) Western blot of SLC7A11 protein levels in *Nrf2* and *SLC7A11* knockdown Hs578T cells. (**b**) Real-time RT-PCR analysis of *Nrf2* and *SLC7A11* mRNA levels in *Nrf2* and *SLC7A11* knockdown MDA-MB-231 cells. Measurements were normalized to 18 s rRNA levels. Data represent the means ± s.d. ($n = 3$); **$P < 0.01$, unpaired Student's *t*-test. (**c,d**) Analysis of *Nrf2* and *SLC7A11* knockdown Hs578T cells. (**c**) Glutamate release into the medium. Data represent the means ± s.d. ($n = 3$); **$P < 0.01$; unpaired Student's *t*-test. (**d**) Cell viability 24 h after glucose withdrawal. EGCG (50 μM) and dm-αKG (4 mM) were added as indicated. Data represent the means ± s.d. ($n = 4$). (**e–g**) MDA-MB-231 cells were treated with vehicle (DMSO) or 15 μM DMF for 24 h in glucose-replete medium. Additional manipulations included knockdown of *SLC7A11* or *Nrf2*, and overexpression of *SLC7A11*. (**e**) Western blot of SLC7A11 levels upon treatment with DMF. (**f**) Glutamate release into media. After DMF-pretreatment, cells were incubated in fresh glucose-deficient medium without DMF, and glutamate released into the medium was measured at 2 h. Data represent the means ± s.d. ($n = 4$). (**g**) Cell viability 24 h after glucose depletion. After DMF-pretreatment, cells were incubated in fresh glucose-deficient medium without DMF for additional 24 h before measurement of cell viability. Data represent the means ± s.d. ($n = 4$); NT, no treatment.

(Supplementary Fig. 6a). Consistent with these pharmacological results, SLC7A11 was upregulated in cybrid cells containing homoplasmic mtDNA mutations in the ND1 subunit of complex I or subunit ATP6 of complex V (NARP) (Fig. 6a)[30,31]. The SLC7A11 induction was particularly marked in the *NARP* cell line, which has been reported to have high levels of ROS and induction of the mitochondrial ROS-scavenging enzyme MnSOD[30]. SLC7A11 induction in the *NARP* cybrid cells was

highly dependent on the Nrf2 and Atf4 transcription factors (Supplementary Fig. 6b) and could be suppressed by treatment with the antioxidant N-acetylcysteine (Supplementary Fig. 6c).

We investigated whether these compensatory changes in system $x_c^-$ expression affect cellular metabolism in the cybrid cells. Consistent with their higher expression of SLC7A11, the *ND1* and *NARP* cybrids had lower intracellular glutamate levels (Fig. 6b). Inhibition of system $x_c^-$ by *SLC7A11* knockdown or a

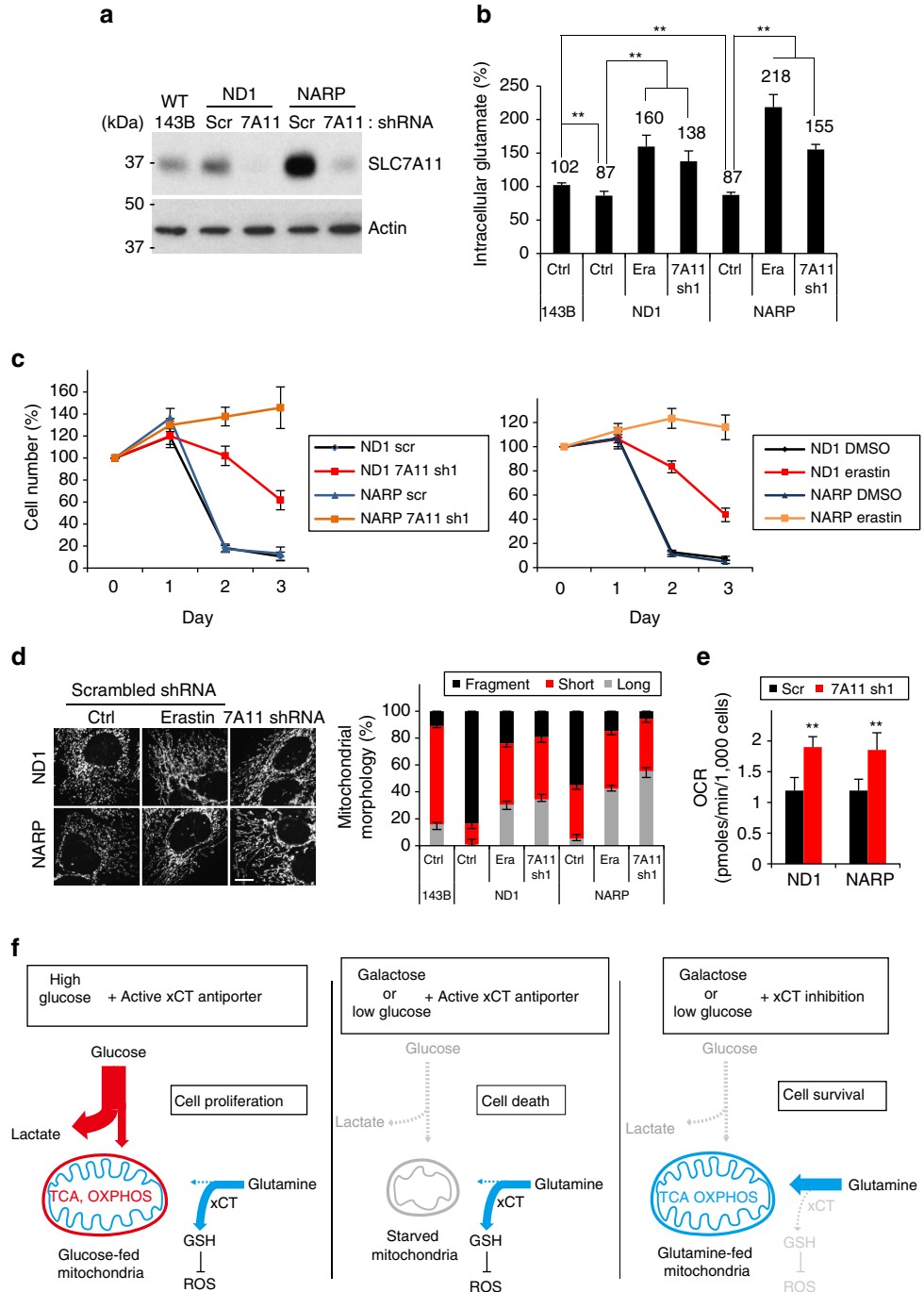

**Figure 6 | Inhibition system x$_c^-$ enhances viability and mitochondrial function of cybrid cells harbouring mtDNA mutations.** (**a**) Western blot of SLC7A11 levels in 143B and isogenic mtDNA-mutant *ND1* and *NARP* cells. (**b**–**e**) Analysis of *SLC7A11* knockdown or erastin (Era)-treated (5 μM) *ND1* and *NARP* cells cultured in the presence of 10 mM galactose and 2 mM glutamine. (**b**) Quantification of intracellular glutamate at 24 h after galactose culture. Data represent the means ± s.d. ($n = 4$); ** $P < 0.01$; unpaired Student's *t*-test. (**c**) Cell viability in galactose medium for 3 days. Data represent the means ± s.d. ($n = 4$). (**d**) Representative images and quantification of mitochondrial morphology at 24 h after galactose culture. Data represent the means ± s.d. ($n = 3$). Scale bar, 10 μm. (**e**) Oxygen consumption was determined at 24 h after galactose culture. Data represent the means ± s.d. ($n = 5$); **$P < 0.01$; unpaired Student's *t*-test. (**f**) Model for the dual effects of the xCT antiporter. The xCT antiporter diverts glutamine metabolism from the TCA cycle into GSH synthesis, which is important for antioxidant function. Under certain cellular conditions, excessive diversion of glutamine from the TCA cycle is detrimental (middle panel) and therefore inhibition of the antiporter improves survival (right panel).

potent inhibitor, erastin, increased intracellular glutamate in *ND1* and *NARP* cells. Similarly, intracellular α-ketoglutarate, a product of glutamate metabolism, was also increased by SLC7A11 inhibition (Supplementary Fig. 6d). It has long been known that cells with OXPHOS defects survive poorly on galactose-containing medium, which forces a higher demand for OXPHOS as the

source of ATP generation[32]. The *ND1* and *NARP*-mutant cybrids, unlike the isogenic WT 143B cells, were not able to survive in galactose-containing medium (Fig. 6c; Supplementary Fig. 6e). However, *SLC7A11* knockdown or erastin treatment strongly rescued the viability of *ND1* and *NARP* cybrids in galactose medium. The effect was most marked in the *NARP*

cybrids, which showed a much greater induction of SLC7A11 (Fig. 6c).

This increase in cell viability was correlated with striking normalization of mitochondrial morphology. Our group previously reported that WT cells elongate their mitochondria under medium conditions that require enhanced OXPHOS activity, whereas cells with mtDNA mutations instead fragment their mitochondria[33]. Consistent with this result, ND1 and NARP cybrids showed marked mitochondrial fragmentation after 24 h of galactose culture. In contrast, cybrids with SLC7A11 knockdown or erastin treatment showed substantial elongation of mitochondria when the carbon source in the medium was switched from glucose to galactose (Fig. 6d; Supplementary Fig. 6f). SLC7A11 knockdown did not change the levels of several mitochondrial dynamics proteins, including Opa1, Mfn1 and Drp1 (Supplementary Fig. 6g). Furthermore, the SLC7A11 knockdown cells showed higher respiration activity than control-mutant cells (Fig. 6e). Taken together, these results suggest that inhibition of system $x_c^-$ in mtDNA-mutant cybrids improves mitochondrial morphology and respiration, resulting in markedly enhanced survival in galactose medium.

## Discussion

System $x_c^-$ is a cystine–glutamate antiporter composed of the light chain subunit xCT/SLC7A11 and the heavy chain subunit SLC3A2/CD98hc. Previous studies have focused on the role of this antiporter in cellular redox homoeostasis, because it imports cystine, the rate-limiting precursor for synthesis of the antioxidant glutathione[19]. The importance of this function is emphasized by the estimate that one-third to one-half of the glutamine taken up by cultured human fibroblasts is used to drive import of cystine[34]. Induction of system $x_c^-$ is thought to be beneficial to cancer cells, because these cells often have elevated levels of ROS. Expression of SLC7A11 in tumour cells has been linked to resistance to anti-cancer drugs[35], induction of tumour growth, cancer cell proliferation, survival[36–38], cell invasion and tumour metastasis[39,40]. As a result, the xCT antiporter has been suggested to be a promising drug target in aggressive breast tumours[38].

However, our studies identify a new aspect of xCT antiporter function. Due to its glutamate export activity, xCT antiporter activity is a key metabolic factor that regulates the nutrient requirements of cultured cells (Fig. 6f). By exporting large amounts of glutamate, an active xCT antiporter system suppresses glutamate-dependent processes in the cell, including mitochondrial respiration. As a result, cells with high system $x_c^-$ activity have limited viability when glucose is removed and glutamine becomes the predominant carbon source. Consistent with this notion, we show in multiple cell lines that xCT antiporter activity is responsible for glucose addiction. When SLC7A11 is disrupted, these cell lines show greatly enhanced survival with glutamine as the major carbon source. This survival depends on conversion of glutamine to glutamate and then α-ketoglutarate, a TCA cycle component. Conversely, overexpression of SLC7A11 is sufficient to make HeLa and SK-BR3 cells highly dependent on glucose versus glutamine.

We show that, through its effect on SLC7A11 levels, Nrf2 is also a factor that can regulate glutamine metabolism and nutrient flexibility in some breast cancer cells. Consistent with our observation that Nrf2 upregulates xCT function, constitutive expression of Nrf2 redirects glutamine flux from the TCA cycle into GSH and NADPH production[41]. We show that high Nrf2 expression limits cells to utilization of glucose, and that removal of Nrf2 enables cells to utilize glutamine more efficiently by maintaining intracellular glutamate levels. These functions are dependent on the Nrf2 target gene SLC7A11.

Why do some tumour cells upregulate Nrf2 and system $x_c^-$, if doing so results in glucose addiction? As a critical component of the antioxidant response, Nrf2 is activated by the higher levels of oxidative stress found in tumour cells. Oncogenes such as Ras, Raf and Myc have been shown to increase Nrf2 transcription as a mode of reducing ROS[42]. In addition, somatic mutations in tumour cells can directly upregulate Nrf2 levels. Normally, Nrf2 levels are kept low by ubiquitination via the E3 ubiquitin ligase adaptor Kelch-like ECH-associated protein 1 (Keap1). Somatic mutations in Nrf2 or Keap1 can abolish Keap1-dependent degradation of Nrf2, resulting in its accumulation and translocation to the nucleus to upregulate antioxidant response genes[43]. Due to its antioxidant effects, Nrf2 expression is critically correlated with tumour growth, drug resistance and poor prognosis in some human cancers[44]. Therefore, we propose the following framework for the dual roles of system $x_c^-$ in antioxidant defence and glutamine metabolism. In the face of high oxidative stress, many cancer cells turn on the Nrf2–SLC7A11 axis to maintain redox homoeostasis. However, a metabolic consequence of high Nrf2–SLC7A11 expression is reduced efficiency of glutamine-dependent processes that drive mitochondrial respiration. This upregulation of system $x_c^-$ contributes to glucose addiction, but is beneficial as long as glucose is plentiful.

Compensatory system $x_c^-$ upregulation also places constraints on the metabolism of cells containing mtDNA mutations. Cybrids with mtDNA mutations, particularly in complex V, show enhanced expression of SLC7A11, likely in response to increased ROS levels[30]. This response has antioxidant benefits but appears to be maladaptive in terms of mitochondrial function. The resulting metabolic rewiring exacerbates the defect in the OXPHOS pathway in cybrids with ND1 and NARP mutations. These homoplasmic cybrids have a 50% reduction in OXPHOS activity[30,31]. They grow in standard high glucose medium but die rapidly when switched to galactose medium, a characteristic phenotype of cells in which glutamine metabolism and OXPHOS is limiting. Remarkably, when SLC7A11 is reduced, these cells are able to survive, and this phenotypic rescue is associated with enhanced mitochondrial respiration and normalization of morphology. These results suggest, in principle, that system $x_c^-$ inhibition may have therapeutic potential in mtDNA disease associated with oxidative stress and maladaptive upregulation of SLC7A11. Such treatments would likely need to be combined with antioxidant supplementation to replace that aspect of system $x_c^-$ function. To test this idea, future studies with animal models of mtDNA diseases will be needed.

## Methods

**Reagents and antibodies.** Dimethyl sulfoxide (DMSO), dm-αKG, dimethyl fumarate (DMF), BPTES, aminooxyacetate (AOA), epigallocatechin gallate (EGCG), erastin, and sulfasalazine (SASP) were obtained from Sigma-Aldrich. Antibodies to the following proteins were used: SLC7A11 (1:2,000, Abcam, ab37185; 1:5,000, Cell Signaling, 12691), SLC3A2 (1:3,000, Santa Cruz BioTech, SC-9160), Hsp60 (1:10,000, Santa Cruz BioTech, sc-1052), Actin (1:10,000, Sigma, A2103). The uncropped scans for key western blots are provided (Supplementary Fig. 7).

**Cell culture.** HeLa (ATCC), 293 T (ATCC), SK-BR-3 (Raymond Deshaies, Caltech), MDA-MB-231 (Raymond Deshaies, Caltech), 143B (Giovanni Manfredi, Weill Medical College), ND1 (Valerio Carelli, University of Bologna), NARP (Giovanni Manfredi, Weill Medical College) and Hs578T (ATCC) cells were cultured in Dulbecco's modified Eagle's medium (DMEM) supplemented with 10% fetal bovine serum. Hap1 (Thijn Brummelkamp, Netherlands Cancer Institute) cells were cultured in Iscove's modified Dulbecco's medium (IMDM) supplemented with 10% fetal bovine serum and additional glutamine. For glucose deprivation, cells were initially incubated overnight in DMEM lacking glucose, pyruvate, and glutamine (Invitrogen, A14430) supplemented with 10% dialysed FBS, 10 mM glucose and 2 mM (breast cancer cells and mtDNA cybrid cells) or

6 mM glutamine (Hap1). The glucose-containing medium was changed to glucose-deficient medium (DMEM A14430 + 10% dialysed FBS + 1 mM glutamine). For oxidative culture, galactose-containing medium was generated by DMEM solution lacking glucose, pyruvate and glutamine supplemented with 10% dialysed FBS, 10 mM galactose and 2 mM glutamine.

**Haploid genetic screen.** Gene-trap vector, pGTpuro, was generated by replacing the GFP gene of pGTen2-ACTB[16] with the puromycin resistance gene. Prior to gene-trap induced random mutagenesis, Hap1 cells were enriched for the haploid karyotype by subcloning. The production of gene-trap virus and mutagenesis of Hap1 cells were performed as previously described[16]. Overall, $10^8$-mutagenized Hap1 cells were incubated in the glucose-deficient medium for 12 days. Cells resistant to glucose depletion were recovered in IMDM medium and formed ~1,200 colonies. Genomic DNA was extracted from the unselected and glucose depletion-selected population, and restricted with MseI (New England Biolabs, R0525) or NlaIII (New England Biolabs, R0125). After self-ligation, genomic DNA was used as template for inverse-PCR. The 5′ primer was:

5′-CAAGCAGAAGACGGCATACGACTGTGTTTCTGTATTTGTCTG-3′ for MseI-digested DNA, or 5′-CAAGCAGAAGACGGCATACGACCCAGGTTAAG ATCAAGGTC-3′ for NlaII-digested DNA. The 3′ primer was:

5′-AATGATACGGCGACCACCGAGATCTACACTCTTTC CCTACACGA CGCTCTTCCGATCTTGCCAAACCTACAGGTGG-3′. Gene-trap insertion sites were identified using inverse-PCR based Illumina HiSeq 2000 sequencing (Illumina, San Diego, CA, USA)[17]. Read alignment to the human hg19 genome and computational comparison of aligned reads between unselected and selected population were performed using Bowtie2, the Integrated Genome Browser and Galaxy, a web-based genome analysis tool[45–47].

**ROS measurement.** Cells were incubated with 10 μM 2′, 7′-dichlorofluorescin diacetate (DCFH-DA) in DMEM lacking phenol red for 30 min at 37 °C and washed with phosphate-buffered saline (PBS). Trypsinized cells were subjected to fluorescence analysis using an Accuri C6 flow cytometer (BD Bioscience).

**Real-time RT-PCR.** Total RNA was extracted from the indicated cells using the SV total RNA isolation kit (Promega) and then reverse transcribed into cDNA using Superscript III First Strand Synthesis kit (Invitrogen). Real-time PCR was performed using the Brilliant III Ultra-Fast SYBR Green QPCR Master Mix (Agilent Technologies) and the CFX96 real-time system (Bio-Rad). Relative fold enrichments of target genes were calculated using the comparative $C_T$ method[48].

**SLC7A11 overexpression.** For SLC7A11 expression, pQCXIP (Clontech) was digested with BamHI, blunt-ended with Klenow polymerase (New England Biolabs), and ligated with the SLC7A11 gene PCR-amplified from a human cDNA library. Cell lines were infected with control retrovirus or retrovirus expressing SLC7A11.

**Stable shRNA.** For shRNA-mediated stable knockdown of SLC7A11 or NRF2, the indicated cell lines were infected with retrovirus expressing shRNA from the human H1 promoter. The targeted sequences were: xCT-Sh1 (5′-GCTGAATTG-GGAACAACTATA-3′), xCT-Sh3 (5′-GCAGTTGCTGGGCTGATTTAT-3′), NRF2-Sh1 (5′-GGTTGAGACTACCATGGTTCC-3′), ATF4-Sh (5′-GCCAAG-CACTTCAAACCTCAT-3′) and scrambled shRNA (5′-CGTTAATCGCGTA-TAATACGC-3′).

**Cell viability under glucose depletion.** Cells were incubated in glucose-deficient medium (DMEM A14430 + 10% dialysed FBS + 1 mM glutamine) for 24 or 48 h. Attached cells at 0, 24 or 48 h were trypsinized, resuspended in fresh medium, and counted with an Accuri C6 flow cytometer (BD Bioscience).

**Immunostaining for mitochondrial morphology.** Cells were fixed in 10% formaldehyde, permeabilized in 0.1% Triton X-100 and immunostained with antibody to HSP60 (1:300, Santa Cruz BioTech). Immunofluorescent images were obtained with a Zeiss LSM 710 confocal microscope (Carl Zeiss). Mitochondrial morphology was scored in triplicates of 100 cells per sample.

**Isotope tracing and mass spectrometry.** For determination of relative oxidative versus reductive TCA flux, steady state isotope labelling of TCA cycle intermediates was employed, as previously described[22,23,25]. Hap1 and HeLa cells with silencing and overexpression, respectively, of xCT were cultured in glucose free media containing 2 mM U-$^{13}C_5$-glutamine (Cambridge Isotope Laboratories) for 8 h. Cells were washed quickly with PBS × 3 and pelleted. Cell pellets were resuspended in 400 μl of ice-cold 80:20 methanol:water, and vortexed for 60 s followed by three freeze-thaw cycles, with alternating 37 °C and − 80 °C liquid baths in 45 s intervals. To ensure complete cell lysis, samples were again vortexed for 30 s and sonicated for 2 min. Samples were then centrifuged at 14,000 $g \times 10$ min at 4 °C, and the supernatants (375 μl) were collected and dried down in vacuo using a vacuum

concentrator. Samples were re-suspended in 80:20 methanol:water according to their cell counts with 50 μl being the volume assigned to the lowest cell count sample with subsequent samples scaled linearly. Resuspended samples were mixed by aspiration, sonicated for 2 min, vortexed for 30 s, centrifuged at 14,000 $g \times 10$ min at 4 °C and supernatants were transferred to LCMS vials containing a 200 μl glass insert. LC–MS/MS-based metabolomics analysis was performed using an Thermo QExactive orbitrap mass spectrometer coupled to a Thermo Vanquish UPLC system. Chromatographic separation of metabolites was achieved using a Phenomenex Luna NH2 aminopropyl column (2.1 × 100 mm, 3 μm) maintained at 35 °C and running a 25 min linear elution gradient starting from 95:5 acetonitrile: 20 mM ammonium acetate pH 9.6–5:95 acetonitrile: 20 mM ammonium acetate pH 9.6, similar to as previously described[49]. An injection volume of 6 μl, which contained metabolic material from ~80,000 cells, was used for all sample injections. The mass spectrometer was operated in negative ion mode running data-dependent MS2 acquisition using a heated electrospray ionization source operated under the following settings: sheath gas flow of 40 U, auxiliary gas flow of 15 U, sweep gas flow of 2 U, spray voltage of − 2.5 kV, capillary temperature of 265 °C, aux gas temp of 350 °C and S-lens RF at 45. For MS1 scan events, a scan range of m/z 67–1,000, mass resolution of 70,000, AGC target at 3e6 and max inject time of 100 ms was used. For MS2 acquisition, a mass resolution of 17.5 k, AGC target at 1e6 and inject time of 50 ms was used. Collected data was imported into the mzMine 2.20 software suite for analysis. Pure standards were used for identification of metabolites through manual inspection of spectral peaks and matching of retention time and MS1 accurate mass, with confirmation of identification through comparison to MS/MS fragmentation patterns.

**Glutamate, glutathione and α-ketoglutarate level assay.** The Amplex Red Glutamate Assay Kit (Molecular Probes), GSH/GSSG Ratio Detection Assay Kit (abcam) and the Alpha-Ketoglutarate Colorimetric/Fluorometric Assay Kit (BioVision) were used for detection of extracellular glutamate released into the medium and intracellular glutamate and α-ketoglutarate in cell lysates. After cells were incubated in fresh glucose-deficient medium for 1 or 2 h, 50 μl medium was taken for extracellular glutamate measurement. For intracellular glutamate and α-ketoglutarate, trypsinized cells were resuspended in 0.1 M Tris-HCl (pH 7.4) and lysed by sonication with three cycles of 10 s on and 30 s off at 20% amplitude. Cell lysates were used for intracellular glutamate, total glutathione and α-ketoglutarate measurement. Values were normalized to protein concentrations of cell lysates.

**OCR and ECAR measurements.** OCR and ECAR of the indicated cell lines were analysed using a Seahorse Bioscience XF96 Extracellular Flux Analyzer (Seahorse Bioscience). Cells were plated into Seahorse tissue culture 96-well plates in DMEM solution lacking glucose, pyruvate, and glutamine (Invitrogen, A14430) supplemented with 10% dialysed FBS, 10 mM glucose or galactose and 2 or 6 mM glutamine and incubated overnight. Prior to measurement, cells were equilibrated in DMEM lacking bicarbonate (Sigma-Aldrich Catalogue #D5030) supplemented with 10% dialysed FBS and the indicated concentration of glutamine ± 10 mM glucose. For OCR and ECAR measurements, four consecutive readings were performed at each time point.

**Statistical analysis.** Statistical significances (P value) of gene-trap insertions on SLC3A2 or SLC7A11 gene were calculated using the one-sided Fisher exact test. Results of cell culture experiments were collected from three or four independent cultures for each sample. Data are presented as means ± s.d. The unpaired Student's t-test was used to calculate P values.

**Data availability.** All data presented in this study are available from the authors upon request.

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

# ARTICLE

8. Choo, A. Y. *et al.* Glucose addiction of TSC null cells is caused by failed mTORC1-dependent balancing of metabolic demand with supply. *Mol. Cell* **38,** 487–499 (2010).

9. DeBerardinis, R. J. *et al.* Beyond aerobic glycolysis: transformed cells can engage in glutamine metabolism that exceeds the requirement for protein and nucleotide synthesis. *Proc. Natl Acad. Sci. USA* **104,** 19345–19350 (2007).

10. Buzzai, M. *et al.* The glucose dependence of Akt-transformed cells can be reversed by pharmacologic activation of fatty acid beta-oxidation. *Oncogene* **24,** 4165–4173 (2005).

11. Yuneva, M., Zamboni, N., Oefner, P., Sachidanandam, R. & Lazebnik, Y. Deficiency in glutamine but not glucose induces MYC-dependent apoptosis in human cells. *J. Cell Biol.* **178,** 93–105 (2007).

12. Wise, D. R. *et al.* Myc regulates a transcriptional program that stimulates mitochondrial glutaminolysis and leads to glutamine addiction. *Proc. Natl Acad. Sci. USA* **105,** 18782–18787 (2008).

13. Schulze, A. & Harris, A. L. How cancer metabolism is tuned for proliferation and vulnerable to disruption. *Nature* **491,** 364–373 (2012).

14. Gohil, V. M. *et al.* Nutrient-sensitized screening for drugs that shift energy metabolism from mitochondrial respiration to glycolysis. *Nat. Biotechnol.* **28,** 249–255 (2010).

15. Carette, J. E. *et al.* Haploid genetic screens in human cells identify host factors used by pathogens. *Science* **326,** 1231–1235 (2009).

16. Jae, L. T. *et al.* Deciphering the glycosylome of dystroglycanopathies using haploid screens for lassa virus entry. *Science* **340,** 479–483 (2013).

17. Carette, J. E. *et al.* Global gene disruption in human cells to assign genes to phenotypes by deep sequencing. *Nat. Biotechnol.* **29,** 542–546 (2011).

18. Palacin, M. *et al.* The genetics of heteromeric amino acid transporters. *Physiology* **20,** 112–124 (2005).

19. Lewerenz, J. *et al.* The cystine/glutamate antiporter system $x_c^-$ in health and disease: from molecular mechanisms to novel therapeutic opportunities. *Antioxid. Redox Signal.* **18,** 522–555 (2013).

20. Gout, P. W., Buckley, A. R., Simms, C. R. & Bruchovsky, N. Sulfasalazine, a potent suppressor of lymphoma growth by inhibition of the $x_c^-$ cystine transporter: a new action for an old drug. *Leukemia* **15,** 1633–1640 (2001).

21. Hensley, C. T., Wasti, A. T. & DeBerardinis, R. J. Glutamine and cancer: cell biology, physiology, and clinical opportunities. *J. Clin. Invest.* **123,** 3678–3684 (2013).

22. Jiang, L. *et al.* Reductive carboxylation supports redox homeostasis during anchorage-independent growth. *Nature* **532,** 255–258 (2016).

23. Metallo, C. M. *et al.* Reductive glutamine metabolism by IDH1 mediates lipogenesis under hypoxia. *Nature* **481,** 380–384 (2012).

24. Mullen, A. R. *et al.* Reductive carboxylation supports growth in tumour cells with defective mitochondria. *Nature* **481,** 385–388 (2012).

25. Zhang, J. *et al.* 13C isotope-assisted methods for quantifying glutamine metabolism in cancer cells. *Methods Enzymol.* **542,** 369–389 (2014).

26. Barretina, J. *et al.* The cancer cell line encyclopedia enables predictive modelling of anticancer drug sensitivity. *Nature* **483,** 603–607 (2012).

27. Hayes, J. D. & Dinkova-Kostova, A. T. The Nrf2 regulatory network provides an interface between redox and intermediary metabolism. *Trends Biochem. Sci.* **39,** 199–218 (2014).

28. Scannevin, R. H. *et al.* Fumarates promote cytoprotection of central nervous system cells against oxidative stress via the nuclear factor (erythroid-derived 2)-like 2 pathway. *J. Pharmacol. Exp. Ther.* **341,** 274–284 (2012).

29. Li, N. *et al.* Mitochondrial complex I inhibitor rotenone induces apoptosis through enhancing mitochondrial reactive oxygen species production. *J. Biol. Chem.* **278,** 8516–8525 (2003).

30. Mattiazzi, M. *et al.* The mtDNA T8993G (NARP) mutation results in an impairment of oxidative phosphorylation that can be improved by antioxidants. *Hum. Mol. Genet.* **13,** 869–879 (2004).

31. Baracca, A. *et al.* Severe impairment of complex I-driven adenosine triphosphate synthesis in leber hereditary optic neuropathy cybrids. *Arch. Neurol.* **62,** 730–736 (2005).

32. Robinson, B. H., Petrova-Benedict, R., Buncic, J. R. & Wallace, D. C. Nonviability of cells with oxidative defects in galactose medium: a screening test for affected patient fibroblasts. *Biochem. Med. Metab. Biol.* **48,** 122–126 (1992).

33. Mishra, P., Carelli, V., Manfredi, G. & Chan, D. C. Proteolytic cleavage of Opa1 stimulates mitochondrial inner membrane fusion and couples fusion to oxidative phosphorylation. *Cell Metab.* **19,** 630–641 (2014).

34. Bannai, S. & Ishii, T. A novel function of glutamine in cell culture: utilization of glutamine for the uptake of cystine in human fibroblasts. *J. Cell Physiol.* **137,** 360–366 (1988).

35. Huang, Y., Dai, Z., Barbacioru, C. & Sadee, W. Cystine-glutamate transporter SLC7A11 in cancer chemosensitivity and chemoresistance. *Cancer Res.* **65,** 7446–7454 (2005).

36. Ishimoto, T. *et al.* CD44 variant regulates redox status in cancer cells by stabilizing the xCT subunit of system $x_c^-$ and thereby promotes tumor growth. *Cancer Cell* **19,** 387–400 (2011).

37. Dixon, S. J. *et al.* Pharmacological inhibition of cystine-glutamate exchange induces endoplasmic reticulum stress and ferroptosis. *eLife* **3,** e02523 (2014).

38. Timmerman, L. A. *et al.* Glutamine sensitivity analysis identifies the xCT antiporter as a common triple-negative breast tumor therapeutic target. *Cancer Cell* **24,** 450–465 (2013).

39. Chen, R. S. *et al.* Disruption of xCT inhibits cancer cell metastasis via the caveolin-1/beta-catenin pathway. *Oncogene* **28,** 599–609 (2009).

40. Yae, T. *et al.* Alternative splicing of CD44 mRNA by ESRP1 enhances lung colonization of metastatic cancer cell. *Nat. Commun.* **3,** 883 (2012).

41. Mitsuishi, Y. *et al.* Nrf2 redirects glucose and glutamine into anabolic pathways in metabolic reprogramming. *Cancer Cell* **22,** 66–79 (2012).

42. DeNicola, G. M. *et al.* Oncogene-induced Nrf2 transcription promotes ROS detoxification and tumorigenesis. *Nature* **475,** 106–109 (2011).

43. Hayes, J. D. & McMahon, M. NRF2 and KEAP1 mutations: permanent activation of an adaptive response in cancer. *Trends Biochem. Sci.* **34,** 176–188 (2009).

44. Moon, E. J. & Giaccia, A. Dual roles of NRF2 in tumor prevention and progression: possible implications in cancer treatment. *Free Radic. Biol. Med.* **79,** 292–299 (2015).

45. Blankenberg, D. *et al.* Galaxy: a web-based genome analysis tool for experimentalists. *Curr. Protoc. Mol. Biol* **89,** 19.10.1–19.10.21 (2010).

46. Langmead, B., Trapnell, C., Pop, M. & Salzberg, S. L. Ultrafast and memory-efficient alignment of short DNA sequences to the human genome. *Genome Biol.* **10,** R25 (2009).

47. Nicol, J. W., Helt, G. A., Blanchard, Jr S. G., Raja, A. & Loraine, A. E. The integrated genome browser: free software for distribution and exploration of genome-scale datasets. *Bioinformatics* **25,** 2730–2731 (2009).

48. Livak, K. J. & Schmittgen, T. D. Analysis of relative gene expression data using real-time quantitative PCR and the $2^{-\Delta\Delta C_T}$ method. *Methods* **25,** 402–408 (2001).

49. Lu, W., Bennett, B. D. & Rabinowitz, J. D. Analytical strategies for LC-MS-based targeted metabolomics. *J. Chromatogr. B Analyt. Technol. Biomed. Life Sci.* **871,** 236–242 (2008).

## Acknowledgements

Hap1 cells were generously provided by Thijn Brummelkamp (Netherlands Cancer Institute), and SK-BR-3 and MDA-MB-231 cells were provided by Raymond Deshaies (Caltech). We also thank Hsiuchen Chen for insightful discussions and comments. This work was supported by a grant from the National Institute of Health (GM110039) to D.C.C., and grants from the Mary Kay Foundation, the V Foundation for Cancer Research and the Sidney Kimmel Foundation to M.J.

## Author contributions

C.-S.S. and D.C.C. conceived the overall project, with contributions from P.M. C.-S.S. performed the majority of the experimental work. P.M. performed some experiments with cybrid cell lines, which were provided by V.C. and M.D'A. J.D.W. and M.J. performed and analysed the mass spectrometry studies. C.-S.S. and D.C.C. wrote the paper, and all authors provided input.

## Additional information

**11**