## [Peer Review File · Nature Communications]

Reviewers' Comments:

Reviewer #1 (Remarks to the Author)

This is a very lucid and exciting paper from one of the top mitochondrial biologists. In this paper, Chan and colleagues demonstrate that the system xCT limits the ability of cells to survive under conditions of low glucose. They began with the observation that a haploid cell line (one typically used for genetic screens) simply cannot survive under low glucose conditions. To explore why, they performed transposon mutagenesis to identify lines that can now survive under low glucose. The results are quite clean and pointed to SLC3A2 or of SLC7A11, which are components of the system xCT, which functions to exchange cytosolic glutamate for extracellular cystine. It has been known that cultured cells derive their ATP either from glycolysis (using glucose as a fuel) or from OXPHOS (using glutamine as a precursor to glutamate). Chan and colleagues suggest that the fate of the glutamine derived glutamate is under the control of system xCT. When active, glutamate is re-directed out of the cell to promote cystine uptake for ROS defense. When ablated, glutamate is available as an anaplerotic input into the TCA cycle. The screen is clever, the follow-up experiments are straightforward and support this novel model. The authors demonstrate the system xCT lies downstream of NRF2/Keap (as expected) and also explore the therapeutic potential of ablating xCT in mitochondrial disease. I think the paper is well written, with the conclusions supported by the data, though I have some qualms about proposing xCT as a therapeutic target for mitochondrial disease.

Major Comments:

1. Stockwell and colleagues have demonstrated that inhibition of xCT leads to ferroptosis. I'm curious to know why these cells do not exhibit ferroptosis. Can the authors speculate as to how do cells in low glucose with ablated xCT produce glutathione? Presumably they cannot rely on de novo cysteine synthesis via trans-sulfuration as it requires glucose-derived serine.
2. One of the authors (Jain) was first author on a 2012 Science paper that performed metabolomics on a large number of cancer cell lines. In that paper's Figure 1B they showed that many of the NCI60 cell lines exhibit glutamine uptake and glutamate release. I do wonder if some of the discordance could be explained by cystine uptake. This could be a nice meta-analysis of that data that could help to support their model.
3. What is the fate/relevance of ammonia that is released in the conversion of glutamine to glutamate?
4. Why aren't there any error bars on the tracer analyses? This seems essential.
5. The idea of xCT as a therapeutic target for mitochondrial disease seems a bit counterintuitive and could even be dangerous on its own. In the paper the authors point out that it would be important to attempt in combination with anti-oxidant therapy. The mitochondrial disease community is desperate for therapies and I would urge the papers to qualify their speculations in the abstract and in the main paper perhaps advocate for animal testing as a next step.

Reviewer #2 (Remarks to the Author)

Shin et al, "The glutamate/cysteine antiporter xCT antagonizes glutamine metabolism and reduces nutrient flexibility"

This work identified xCT cystine/glutamate antiporter as a limiting factor of metabolic flexibility, conferring glucose addiction phenotype on cells. The authors conducted a haploid genetic screen

using Hap1 cells and identified factors involved in the glucose addiction and obtained two subunits of xCT, SLC7A11 and SLC3A2. The authors modulated xCT activity by knocking down and overexpressing SLC7A11, which is a specific subunit of xCT, and by knocking down NRF2, which is a transcriptional activator of SLC7A11 gene. Decrease of xCT activity improved cell viability under glucose deprivation, whereas increase of xCT activity sensitized cells to glucose deprivation. Similar effects of xCT on cellular metabolism and cell viability were observed in breast cancer cell lines. Finally the authors examined cybrid cells harboring mutations in mitochondrial DNA, which exhibit maladaptive upregulation of xCT, and found that xCT inhibition improved mitochondrial morphology and OXPHOS function that were impaired in galactose medium. This work is very well organized with high-quality data and has described a new concept that xCT, which has been regarded as a critical transporter for the uptake of cystine and glutathione synthesis, contributes to the glucose addiction by limiting utilization of glutamine. However, considering that xCT is an antiporter of cystine and glutamate, the results described in the manuscript are theoretically well expected. An important question is how this aspect of xCT, limiting the utilization of glutamate, is critical within the physiological range of glucose and glutamine availability in vivo. Several questions below need to be addressed.

Major points;

1) Completely no glucose condition is hard to be encountered in vivo. In addition to the "glucose withdrawal" (0 mg/l glucose) condition, effects of high and low levels of xCT on cell viability and mitochondrial function in low-glucose conditions, such as 10 mg/l and 100 mg/l glucose, would be examined.

2) In hypoxic condition, glutamine is an important carbon source for fatty acid synthesis and cell growth via reductive carboxylation. According to the authors' argument, xCT-overexpressing cells are expected to be vulnerable to hypoxic condition due to inefficient glutamine utilization. Effects of high and low levels of xCT on cell proliferation would be examined under hypoxic condition with physiological levels of glucose and glutamine.

3) NRF2 activity is mainly regulated at its protein level rather than its mRNA level. When the correlation between NRF2 activity and SLC7A11 expression is examined, mRNA levels of NRF2 target genes (e.g. NQO1, GCLC, GCLM, TXNRD1, etc.) are better to be examined instead of NRF2 mRNA level, which should give much stronger correlation.

4) The relation between mitochondrial function and morphology is not very clear. It is known that mitochondrial morphology is regulated by Drp1, Mfn2 and several other proteins. Does glutamate availability influence the expression or activity of these factors that regulate the mitochondrial morphology?

Minor point;

1) A rationale for using galactose medium in the experiments shown in Figure 6, but not glucose-deficient medium, would be explained.

Reviewer #3 (Remarks to the Author)

The authors have demonstrated that xCT plays a role in enhancing cellular glucose dependence. This is a very interesting aspect as a function of xCT. The study is well designed and performed, however, the reviewer has some concerns and suggestions.

Major points:

1. In Hap1 cells, are there any glutamate transport systems other than system xc-, for example, system XAG-? In addition, do parental Hap1 cells die when they are cultured in the glucose-containing normal medium with sulfasalazine for a few days? These are important points to show that the glutamate transport is exclusively mediated by system xc- in these cells.

2. How do the intracellular glutathione change after removal of glucose and modulating xCT expression in WT Hap1 or HeLa cells? Glutathione may affect the observed phenomenon, thus, the data on the change of intracellular glutathione should be shown.

3. If glutathione synthesis is inhibited by the inhibitor such as buthionine sulfoximine, will the rescued cell death caused by disruption of xCT function after glucose withdrawal be cancelled in Hap1 or Hs578T cells?

4. In Figs. 3a and 3b, how about the glutamine transport activity in these cells? Intracellular glutamate levels may be affected not only by system xc⁻ but also by the neutral amino acid transport systems which mediates glutamine transport. Similarly, in Supplementary Fig. 5b, how about the glutamine transport activity in Hs578T and SK-BR3 cells?

5. In Methods, the authors mention that "cells were initially cultured and 2 or 6 mM glutamine". Why and in which experiments did the authors use 2 mM and 6 mM glutamine containing medium? In addition, in glucose-deficient medium, 1 mM glutamine is added. Why did the authors decrease the concentration of glutamine in glucose-deficient medium?

6. In Fig. 3d (Hap1 cells) and Fig. 4d (Hs578T cells), the effect of BPTES seems to be different among both types of the cells. Why does this happen?

7. In Fig. 4d, the extent of rescued cells by SASP is much lower than that by shRNA treatment. Judging from the concentration of SASP used in this experiment, the activity of system xc⁻ must be almost completely inhibited. Why does this happen?

Minor points:

1. Through all over the text, "system Xc⁻" should be "system xc⁻". Capital letters generally means sodium-dependent transport systems except system L.

2. The authors seem to confuse the words, system xc⁻, xCT, and SLC7A11. System xc⁻ consists of xCT and 4F2hc. In other words, system xc⁻ consists of SLC7A11 and SLC3A2. Thus, xCT and SLC7A11 are the same, and not equal to system xc⁻. To avoid the confusion, xCT or SLC7A11 should be unified in the text. The reviewer feels that xCT should be used to express the transporter protein.

3. It may be helpful for readers that a schematic figure such as Supplementary Fig. 7 is added into the main text.

4. It is important to show where the antibody to xCT was obtained from, because many anti-xCT antibodies are not specific to xCT, even if commercially available ones.

We thank the reviewers for the helpful comments. Below is a point-by-point response to the reviewers' comments.

Reviewer #1 (Remarks to the Author):

This is a very lucid and exciting paper from one of the top mitochondrial biologists. In this paper, Chan and colleagues demonstrate that the system xCT limits the ability of cells to survive under conditions of low glucose. They began with the observation that a haploid cell line (one typically used for genetic screens) simply cannot survive under low glucose conditions. To explore why, they performed transposon mutagenesis to identify lines that can now survive under low glucose. The results are quite clean and pointed to SLC3A2 or of SLC7A11, which are components of the system xCT, which functions to exchange cytosolic glutamate for extracellular cystine. It has been known that cultured cells derive their ATP either from glycolysis (using glucose as a fuel) or from OXPHOS (using glutamine as a precursor to glutamate). Chan and colleagues suggest that the fate of the glutamine derived glutamate is under the control of system xCT. When active, glutamate is re-directed out of the cell to promote cystine uptake for ROS defense. When ablated, glutamate is available as an anaplerotic input into the TCA cycle. The screen is clever, the follow-up experiments are straightforward and support this novel model. The authors demonstrate the system xCT lies downstream of NRF2/Keap (as expected) and also explore the therapeutic potential of ablating xCT in mitochondrial disease. I think the paper is well written, with the conclusions supported by the data, though I have some qualms about proposing xCT as a therapeutic target for mitochondrial disease.

Response: We thank the reviewer for the enthusiastic comments.

Major Comments:

1. Stockwell and colleagues have demonstrated that inhibition of xCT leads to ferroptosis. I'm curious to know why these cells do not exhibit ferroptosis. Can the authors speculate as to how do cells in low glucose with ablated xCT produce glutathione? Presumably they cannot rely on de novo cysteine synthesis via trans-sulfuration as it requires glucose-derived serine.

Response: As noted by the reviewer, Stockwell and colleagues (Cell (2012)149:1060) found that erastin causes a form of cell death termed ferroptosis in a subset of cancer cells. In such cases, "overwhelming, iron-dependent accumulation of ROS" cause cell death.

In the case of the Hap1 and Hs578T cells, inhibition of xCT does not cause cell death and in fact improves cell viability under low glucose conditions. In these cell models, presumably the levels of ROS are not sufficient to cause cell death, even though the levels of glutathione are indeed reduced when SLC7A11 is depleted (please see point #2 to Reviewer 3). Under low glucose conditions, the benefits of glutamate availability for the TCA cycle outweigh the reduction in anti-oxidant activity.

The reviewer brings up the idea of whether trans-sulfuration might help to maintain glutathione in the absence of SLC7A11. In principle, this seems possible because DMEM culture media contains serine. In the absence of glucose-derived serine, serine imported from the culture medium may be sufficient to maintain the trans-sulfuration pathway.

2. One of the authors (Jain) was first author on a 2012 Science paper that performed metabolomics on a large number of cancer cell lines. In that paper's Figure 1B they showed that

many of the NCI60 cell lines exhibit glutamine uptake and glutamate release. I do wonder if some of the discordance could be explained by cystine uptake. This could be a nice meta-analysis of that data that could help to support their model.

Response: This is an interesting suggestion, and we have discussed the issue with our co-author Mohit Jain. He notes that the mass spectrometry analysis performed in his 2012 Science paper was targeted using ion-pairing and hydrophobic interaction liquid chromatography. Cysteine is poorly retained by these chromatography methods, and therefore neither cysteine nor cystine was measured in those datasets.

3. What is the fate/relevance of ammonia that is released in the conversion of glutamine to glutamate?

Response: Based on the literature, ammonia generated by conversion of glutamine to glutamate and alpha-ketoglutarate is secreted into the culture medium and unavailable to produce biomass.

4. Why aren't there any error bars on the tracer analyses? This seems essential.

Response: This is a valid point, and we have repeated the flux tracing analysis in triplicate. The new plots in Fig. 3g and Supplementary Fig. 3g now have error bars, and support our conclusion that reductive carboxylation is not enhanced in our experimental treatments.

5. The idea of xCT as a therapeutic target for mitochondrial disease seems a bit counterintuitive and could even be dangerous on its own. In the paper the authors point out that it would be important to attempt in combination with anti-oxidant therapy. The mitochondrial disease community is desperate for therapies and I would urge the papers to qualify their speculations in the abstract and in the main paper perhaps advocate for animal testing as a next step.

Response: We agree with the caveats noted by the reviewer and have re-written the Abstract and Discussion to give a more balanced presentation.

Reviewer #2 (Remarks to the Author):

Shin et al, "The glutamate/cysteine antiporter xCT antagonizes glutamine metabolism and reduces nutrient flexibility"

This work identified xCT cystine/glutamate antiporter as a limiting factor of metabolic flexibility, conferring glucose addiction phenotype on cells. The authors conducted a haploid genetic screen using Hap1 cells and identified factors involved in the glucose addiction and obtained two subunits of xCT, SLC7A11 and SLC3A2. The authors modulated xCT activity by knocking down and overexpressing SLC7A11, which is a specific subunit of xCT, and by knocking down NRF2, which is a transcriptional activator of SLC7A11 gene. Decrease of xCT activity improved cell viability under glucose deprivation, whereas increase of xCT activity sensitized cells to glucose deprivation. Similar effects of xCT on cellular metabolism and cell viability were observed in breast cancer cell lines. Finally the authors examined cybrid cells harboring mutations in mitochondrial DNA, which exhibit maladaptive upregulation of xCT, and found that xCT inhibition improved mitochondrial morphology and OXPHOS function that were impaired in galactose medium.

This work is very well organized with high-quality data and has described a new concept that xCT, which has been regarded as a critical transporter for the uptake of cystine and glutathione

synthesis, contributes to the glucose addiction by limiting utilization of glutamine. However, considering that xCT is an antiporter of cystine and glutamate, the results described in the manuscript are theoretically well expected. An important question is how this aspect of xCT, limiting the utilization of glutamate, is critical within the physiological range of glucose and glutamine availability in vivo. Several questions below need to be addressed.

Major points;

1) Completely no glucose condition is hard to be encountered in vivo. In addition to the “glucose withdrawal” (0 mg/l glucose) condition, effects of high and low levels of xCT on cell viability and mitochondrial function in low-glucose conditions, such as 10 mg/l and 100 mg/l glucose, would be examined.

Response: To examine this issue, we tested Hap1 and HeLa viability and respiration in a low-glucose condition (0.2 mM glucose, equivalent to 36 mg /L). The results show the same trend as the no glucose experiments. Knockdown of SLC7A11 in Hap1 cells resulted in much higher cell viability in low-glucose culture. Conversely, overexpression of SLC7A11 in HeLa cells inhibited cell growth. Control and SLC7A11 knockdown Hap1 cells show comparable oxygen consumption after a short period culture (8 hr) in low glucose. In contrast, SLC7A11-overexpressing HeLa cells showed lower respiration than control cells. The viability data has been added to the revised manuscript as Supplementary Fig. 2b.

2) In hypoxic condition, glutamine is an important carbon source for fatty acid synthesis and cell growth via reductive carboxylation. According to the authors' argument, xCT-overexpressing cells are expected to be vulnerable to hypoxic condition due to inefficient glutamine utilization. Effects of high and low levels of xCT on cell proliferation would be examined under hypoxic condition with physiological levels of glucose and glutamine.

Response: Using Hap1 and HeLa cells, we tested the effect of xCT knockdown and overexpression in cells grown under hypoxic conditions (1% O₂). Cells were cultured with 4 mM glucose and 1 mM glutamine. In Hap1 cells under hypoxic conditions, SLC7A11 knockdown had no effect on the growth rate. With HeLa cells, the growth rate was substantially reduced with hypoxia; however, this effect was not changed with SLC7A11 overexpression.

3) NRF2 activity is mainly regulated at its protein level rather than its mRNA level. When the correlation between NRF2 activity and SLC7A11 expression is examined, mRNA levels of NRF2 target genes (e.g. NQO1, GCLC, GCLM, TXNRD1, etc.) are better to be examined instead of NRF2 mRNA level, which should give much stronger correlation.

Response: This is an excellent point. Consistent with the reviewer's comment, SLC7A11 shows high correlation with multiple NRF2 target genes across 947 cancer cell lines. The correlation coefficient (R) between SLC7A11 and NRF2 target genes are: NFE2L2 (+0.4306), NQO1 (+0.4179), GCLC (+0.3283), GCLM (+0.3944), TXNRD1 (+0.4321). This information has been added to the revised text.

4) The relation between mitochondrial function and morphology is not very clear. It is known that mitochondrial morphology is regulated by Drp1, Mfn2 and several other proteins. Does glutamate availability influence the expression or activity of these factors that regulate the mitochondrial morphology?

Response: We tested the levels of several mitochondrial dynamics proteins under glucose and galactose conditions. We did not see an obvious difference in their steady-state protein levels as a function of 7A11. We have add the Western blot result of mtDNA mutant cells in Supplementary Fig. 6g.

Minor point;

1) A rationale for using galactose medium in the experiments shown in Figure 6, but not glucose-deficient medium, would be explained.

Response: This is a good point, and we have provided the following explanation in the text. Galactose-containing medium, which requires cells to generate more of their ATP via OXPHOS,

has long been used to study cells with mtDNA mutations.

Reviewer #3 (Remarks to the Author):

The authors have demonstrated that xCT plays a role in enhancing cellular glucose dependence. This is a very interesting aspect as a function of xCT. The study is well designed and performed, however, the reviewer has some concerns and suggestions.

Response: We thank the reviewer for the supportive comments.

Major points:

1. In Hap1 cells, are there any glutamate transport systems other than system xc-, for example, system XAG-? In addition, do parental Hap1 cells die when they are cultured in the glucose-containing normal medium with sulfasalazine for a few days? These are important points to show that the glutamate transport is exclusively mediated by system xc- in these cells.

Response: We think the Hap1 cells have additional glutamate transporters other than system xc-. Our SLC7A11 shRNAs work quite efficiently (> 90% reduction in the SLC7A11 protein) in Hap1 and Hs578T cells (Fig 2a and 4a). The SLC7A11 knockdown cells showed reduced, but still substantial, glutamate release: 30% with Hap1 (Fig 2b) and 60% for Hs578T (Supplementary Fig 5a). These results indicate that Hap1 and Hs578T cells have additional glutamate transporters beyond system xc-. With 10 mM glucose and 2 mM glutamine, Hap1 and Hs578T cells grow normally even under 500 uM sulfasalazine treatment.

2. How do the intracellular glutathione change after removal of glucose and modulating xCT expression in WT Hap1 or HeLa cells? Glutathione may affect the observed phenomenon, thus, the data on the change of intracellular glutathione should be shown.

Response: To address this issue, we measured the intracellular GSH level 1 hr after glucose withdrawal. SLC7A11 knockdown decreased the GSH level in Hap1 cells. Conversely, SLC7A11 overexpression in HeLa cells increased the GSH level. These results suggest that even though SLC7A11 inhibition is beneficial for cell viability after glucose withdrawal, it does so at the cost of decreased glutathione. We added this result to Supplementary Fig. 3b.

Intracellular glutathione level at 1 hr of glucose depletion (%)

3. If glutathione synthesis is inhibited by the inhibitor such as buthionine sulfoximine, will the rescued cell death caused by disruption of xCT function after glucose withdrawal be cancelled in Hap1 or Hs578T cells?

Response: To address this issue, we measured the effect of buthionine sulfoximine. Knockdown of SLC7A11 increased Hap1 viability upon glucose withdrawal even in the presence of buthionine sulfoximine. This results again argues that glutathione levels are not critical for Hap1

cells under our culture conditions, and that the metabolic effects of SLC7A11 inhibition are more important.

4. In Figs. 3a and 3b, how about the glutamine transport activity in these cells? Intracellular glutamate levels may be affected not only by system xc- but also by the neutral amino acid transport systems which mediates glutamine transport. Similarly, in Supplementary Fig. 5b, how about the glutamine transport activity in Hs578T and SK-BR3 cells?

Response: To monitor glutamine transport, we measured the residual glutamine level in the culture medium 8 hr after glucose depletion. The baseline fresh medium contains no glucose and 2 mM glutamine. The results below show that neither SLC7A11 knockdown nor overexpression significantly changed glutamine consumption of cells during glucose depletion. These results imply that a decrease or increase in SLC7A11 does not affect the activity of other glutamine transporters.

Glutamine level in glucose-depleted medium at 8 hr (%)

5. In Methods, the authors mention that “cells were initially cultured and 2 or 6 mM glutamine”. Why and in which experiments did the authors use 2 mM and 6 mM glutamine containing medium? In addition, in glucose-deficient medium, 1 mM glutamine is added. Why did the authors decrease the concentration of glutamine in glucose-deficient medium?

Response: We have revised the Methods to clarify the concentrations used. We used 6 mM glutamine for Hap1 cells and 2 mM glutamine for breast cancer cells and the mtDNA cybrid cells. These concentrations were determined to be optimal for cell growth of the various cell lines. For glucose depletion, we standardized the glutamine concentration to 1 mM, but the results were similar at a range of glutamine concentrations (Supplementary Fig 3e).

6. In Fig. 3d (Hap1 cells) and Fig. 4d (Hs578T cells), the effect of BPTES seems to be different among both types of the cells. Why does this happen?

Response: This is an interesting point that we addressed partially in the original text. We have now provided a more explicit discussion of this issue. BPTES inhibits a specific isoform (GLS1) of glutaminase. Based on our results, GLS1 appears to be the primary glutaminase in Hs578T cells but not Hap1 cells.

7. In Fig. 4d, the extent of rescued cells by SASP is much lower than that by shRNA treatment. Judging from the concentration of SASP used in this experiment, the activity of system xc- must be almost completely inhibited. Why does this happen?

Response: This is an interesting observation. We were also curious why shRNA knockdown showed more efficient rescue than SASP. We can propose two possibilities. First, the consequence of SASP treatment and shRNA are inherently somewhat different. SASP treatment cause acute inactivation, whereas shRNA occurs over a long time frame and may allow adaptive changes. Second, it is possible that the two treatments affect system xc- to different extents. Perhaps the shRNA treatment allows a residual level of system xc- activity that is beneficial (by allowing residual glutathione production) for cell viability during glucose withdrawal.

Minor points:

1. Through all over the text, "system Xc-" should be "system xc-". Capital letters generally means sodium-dependent transport systems except system L.

Response: Thank you for pointing out this issue. We have gone through the text and corrected the nomenclature as suggested.

2. The authors seem to confuse the words, system xc-, xCT, and SLC7A11. System xc- consists of xCT and 4F2hc. In other words, system xc- consists of SLC7A11 and SLC3A2. Thus, xCT and SLC7A11 are the same, and not equal to system xc-. To avoid the confusion, xCT or SLC7A11 should be unified in the text. The reviewer feels that xCT should be used to express the transporter protein.

Response: We agree with the reviewer that it is important to be careful with the nomenclature. We have gone through the paper to make sure the following convention is consistently used. The terms "system xc-" and "xCT antiporter" are used to indicate the functional antiporter complex. When indicating the protein subunit or gene, we use the terms "SLC7A11" or "xCT subunit".

3. It may be helpful for readers that a schematic figure such as Supplementary Fig. 7 is added into the main text.

Response: As suggested, we have moved this figure to the main text, Fig. 6f.

4. It is important to show where the antibody to xCT was obtained from, because many anti-xCT antibodies are not specific to xCT, even if commercially available ones.

Response: The revised manuscript now contains the source and catalog number of all antibodies used.

Reviewers' Comments:

Reviewer #1 (Remarks to the Author)

I think the revised manuscript is suitable for publication.

Reviewer #2 (Remarks to the Author)

The questions from this reviewer have been almost appropriately answered.

Reviewer #3 (Remarks to the Author)

The authors have properly revised the manuscript, according as the reviewers' comments.

Response to reviewers:

The reviewers did not request any additional data.